# Modular Design in Triboelectric Sensors: A Review on the Clinical Applications for Real-Time Diagnosis

**DOI:** 10.3390/s23094194

**Published:** 2023-04-22

**Authors:** Zequan Zhao, Yin Lu, Yajun Mi, Qiliang Zhu, Jiajing Meng, Xueqing Wang, Xia Cao, Ning Wang

**Affiliations:** 1Center for Green Innovation, School of Mathematics and Physics, University of Science and Technology Beijing, Beijing 100083, China; 2Beijing Institute of Nanoenergy and Nanosystems, Chinese Academy of Sciences, Beijing 100083, China; 3School of Chemistry and Biological Engineering, University of Science and Technology Beijing, Beijing 100083, China

**Keywords:** modular design, real-time diagnosis, triboelectric sensor, self-powered system

## Abstract

Triboelectric nanogenerators (TENGs) have garnered considerable interest as a promising technology for energy harvesting and stimulus sensing. While TENGs facilitate the generation of electricity from micro-motions, the modular design of TENG-based modular sensing systems (TMSs) also offers significant potential for powering biosensors and other medical devices, thus reducing dependence on external power sources and enabling biological processes to be monitored in real time. Moreover, TENGs can be customised and personalized to address individual patient needs while ensuring biocompatibility and safety, ultimately enhancing the efficiency and security of diagnosis and treatment. In this review, we concentrate on recent advancements in the modular design of TMSs for clinical applications with an emphasis on their potential for personalised real-time diagnosis. We also examine the design and fabrication of TMSs, their sensitivity and specificity, and their capabilities of detecting biomarkers for disease diagnosis and monitoring. Furthermore, we investigate the application of TENGs to energy harvesting and real-time monitoring in wearable and implantable medical devices, underscore the promising prospects of personalised and modular TMSs in advancing real-time diagnosis for clinical applications, and offer insights into the future direction of this burgeoning field.

## 1. Introduction

Since its invention in 2012, the TENG has gained considerable attention for its potential to harvest energy and serve as an active sensor in diverse fields, such as green energy, molecular detection, and healthcare [1]. TENGs can convert mechanical energy into electrical energy, making them a promising technology for energy-harvesting applications. Moreover, their ability to generate electrical signals in response to various stimuli, such as mechanical pressure, vibration, temperature changes, and position, makes TENGs potential sensors for multiple applications [2,3].

Meanwhile, as the Internet of Things (IoT) continues to evolve rapidly, TENG-based energy-harvesting and environment-sensing technologies have garnered further interest due to the unique advantages of self-power supply and miniaturisation [4,5,6]. This makes growing innovation in personalised diagnosis possible, which is especially attractive if supported by information technologies such as wireless connectivity, cloud services, and data storage [7]. TMSs can seamlessly integrate with IoT devices, facilitating real-time data collection, processing, and sharing for remote monitoring and analysis. This integration can significantly improve the accuracy and efficiency of medical diagnosis and treatment, ultimately leading to better patient outcomes.

Theoretically, the critical advantage of TENG technology is its compatibility with a wide range of raw materials, which has led to the development of TENG-based biosensors with significant potential for real-time diagnosis in clinical applications due to the wide selection in triboelectric polarity, biocompatibility, and mechanical and other fascinating properties [8,9]. For instance, TENGs can be used to measure various clinical parameters, such as glucose levels, blood pressure, heart rate, and respiratory rate [10,11]. These biosensors can be conveniently worn as a patch or integrated into wearable devices, allowing the continuous monitoring and analysis of physiological signals to be achieved [12].

Modular design in TMSs offers numerous advantages for clinical applications. Firstly, processing using a sensor module, TENGs demonstrate excellent sensitivity, allowing the subtle changes in physical and chemical signals to be detected [13]. Secondly, selecting highly biocompatible modules makes them safe for long-term implantation in the human body. Thirdly, the TENG power generation module can harvest energy from the body, enabling them to operate continuously without the need for external power sources [14]. Lastly, this modular design can be customised to specific patient needs, providing accurate and reliable real-time diagnosis and treatment [15]. These benefits underscore the enormous potential of modular design in TMSs for clinical applications in real-time diagnosis, making them ideal candidates for healthcare applications [16,17,18].

In this review, we aim to provide a comprehensive overview of TMSs and their applications in healthcare, focusing on the role of modularity in the design of such biosensing systems (Figure 1). We will discuss the underlying principles of TENGs, as well as their fabrication methods and material choices. Additionally, we will explore the current state of modular TMSs in various clinical applications and highlight their advantages and challenges for their future development.

## 2. Biological Energy Collection

TENG-based devices promise to support intricate functional modules tailored to address the distinct needs of diverse user groups. To fully realise this potential, TENGs must deliver higher power output. This can be accomplished by expanding the approaches employed in biological energy harvesting, ultimately paving the way for more effective solutions [19,20].

### 2.1. Working Principle of TENGs

TENGs utilise the triboelectric effect and electrostatic induction coupling to convert biomechanical energy into electrical energy [21,22]. When two materials with different degrees of electronegativity come into contact, electrons flow between them [23,24]. When they are separated, electrostatic induction causes electrons to flow to the external load, generating alternating currents by repeating the contact–separation cycles [25,26].

Working modes of TENGs: TENGs can be categorized into four working modes: vertical contact–separation mode, lateral sliding mode, single-electrode mode, and independent triboelectric layer mode (Figure 2).

Vertical contact–separation (CS) mode: Electrons are exchanged at the contact surface when two objects with different degrees of electronegativity are in vertical contact. The separation of the two objects causes electrostatic induction, leading to a potential difference between the electrodes attached to them, generating electric current. With the repetition of contact–separation cycles, alternating current is output [27].

Lateral sliding (LS) mode: Similar to the vertical contact–separation mode, the principle of lateral sliding involves the horizontal displacement of the two objects. An alternating current is generated by repeated removal in the horizontal direction [28].

Single-electrode (SE) mode: This mode uses the Earth as an electrode to generate a potential difference between the metal electrode and the Earth with electrostatic induction, producing current [29].

Freestanding triboelectric layer (FT) mode: The FT mode involves placing a charged object between two electrodes attached to the dielectric layer. The movement of the charged object between the two electrodes changes the potential difference between them, generating current [30].

TENGs can be designed according to the personalized needs of patients, utilizing a variety of friction layer materials and electrode materials. With the ability to generate high-power output, TENG-based devices have the potential to support complex functional modules, making them suitable for a wide range of medical applications.

### 2.2. Motion-Based TENGs

Motion-based TENGs represent cutting-edge development in the field of TENG technology. By converting mechanical energy into electrical energy, these innovative devices are particularly suited for wearable and self-powered sensing applications. Modularity is vital in motion-based TENGs, as it facilitates customisation and adaptability for various use cases and environments (Table 1). This feature dramatically enhances the devices’ efficiency and versatility [31].

Saqib et al. introduced a novel TENG known as the triboelectric particle nanogenerator (P-TENG), which utilises rapidly degradable gelatine capsules and cellulose-based particles to harvest energy from all directions without the need for traditional contact and separation (Figure 3a) [32]. The P-TENG generates voltage ranging from 15 to 85 V and power from 5.488 to 70 μW, achieving the maximum energy conversion efficiency of 74.35%. The modular and lightweight design of the P-TENG makes it suitable for capturing energy from small and irregular movements. Moreover, multiple devices can be connected to enhance electrical energy scavenging. By incorporating a rapidly degradable gelatine capsule and cellulose-based TENG, the P-TENG presents an eco-friendly solution for energy harvesting.

In separate development research, Zhang et al. proposed a modular approach for creating a TENG device capable of harvesting mechanical energy from various human body parts (Figure 3b) [33]. This device serves both energy-harvesting and self-powered sensing purposes. The design features three-dimensional polypyrrole nanoarrays (3D PPy NAs) and porous poly(vinylidene fluoride) (PVDF) films. The unique micro–nanostructures of the PPy NAs enhance the TENG’s effective contact area and contact affinity, leading to improved frictional initiation effects. Additionally, the PVDF films can be tailored to optimize the TENG’s performance in collecting mechanical energy from specific body parts, such as hands, feet, and armpits. The modular design of this device achieves flexibility in adapting to different applications and environments, making it a versatile solution for wearable devices and self-powered sensing.

### 2.3. Implantable TENGs

Implantable TENGs capitalise on innovative technology to develop versatile, self-powered medical devices seamlessly implanted within the human body. These devices are specifically designed to target various health conditions, fostering healing and promoting tissue maturation.

In one study, Zhao et al. developed an advanced, self-powered implantable electrical stimulator utilising a TENG to address damaged myocardium (Figure 3c) [34]. This innovative device generates an electric field on interdigitated electrodes, thereby facilitating the maturation of neonatal rat cardiomyocytes by enhancing the expression of specific proteins. Furthermore, it improves sarcomere organisation, fracture formation, and intracellular calcium levels. Driven by physiological movement, the TENG functions as an implantable medical electronic device for electrically promoting cardiomyocyte maturation, offering invaluable technical support for treating myocardial defects and restoring cardiac tissue function.

In a separate study, Kim et al. engineered a TENG using a biocompatible and biodegradable hydrogel film composed of hyaluronic acid (HA) [35]. This groundbreaking HA-TENG can convert mechanical energy into electricity, thus serving as a renewable energy source for bio-implantable devices (Figure 3d). The device exhibits impressive power output, long-term stability, and minimal cytotoxicity, rendering it suitable for tissue engineering and other biomedical applications. Manufactured with both pure HA and cross-linked HA hydrogel film, the HA-TENG consistently generates power during prolonged testing. The successful implementation of HA-TENGs offers a persuasive proof of concept for future biodegradable and implantable devices, opening new avenues for transient self-powered systems in a wide range of biomedical applications.

## 3. Modular Design in Respiratory and Cardiovascular Systems

### 3.1. Respiratory System

#### 3.1.1. TMSs for Gas Diagnosis

The TENG-based gas diagnostic modular system represents a groundbreaking approach of TENG-based biosensors to the real-time diagnosis of respiratory gases in humans [36,37]. As a critical aspect of this system, modularity enables customisation and adaptability to be achieved, thereby enhancing the performance and usability of TMSs across a range of diagnostic applications (Table 2).

Wang et al. have developed a multifunctional self-powered detection system that integrates four TENGs and a gas sensor, facilitating the real-time diagnosis of respiratory gases [38]. This system comprises three distinct modules: a wind-driven TENG made of PVA/Ag nanofibers and FEP film, a voltage regulator module, and a Ti_3_C_2_Tx MXene/WO_3_-based NO_2_ sensor (Figure 4a). The PVA/Ag nanofiber-based TENG is specifically designed for monitoring harmful respiratory gases, optimized using electrospinning technology to produce the PVA/Ag nanofiber film. Capable of generating open-circuit voltage of up to 530 V and power density of up to 359 mW/m^2^ at 8 m/s wind speed, the TENG demonstrates its efficiency. Moreover, the Ti_3_C_2_Tx MXene/WO_3_-based NO_2_ sensor, powered by the TENG, exhibits excellent responsiveness to NO_2_ gas at room temperature, showcasing its potential in real-time human respiration diagnosis.

In complementary development research, Zhang and his team designed a self-powered sensing system that employs a TENG to detect exhaled gas and diagnose diseases [39]. What sets this system apart is the TENG’s ability to be driven by respiration, functioning as both a power source and a sensor (Figure 4b). The TENG comprises a Ti_3_C_2_Tx MXene/NH_2_-MWCNT composite, and both the friction layer and electrode. The system’s modular design encompasses the TENG, the MXene/NH_2_-MWCNT composite, and the support vector machine model for respiratory type identification. Consequently, the device boasts excellent gas sensing response, a low detection limit, and a rapid response/recovery time. The TENG also holds potential application value in diagnosing diseases related to exhaled gas and can differentiate various respiratory types using a support vector machine model.

The TENG modular gas diagnostic system epitomises a state-of-the-art solution for real-time respiratory gas analysis, leveraging sophisticated materials and modularity to augment functionality and cater to a wide range of clinical applications. The advancements made by Wang et al. and Zhang’s team emphasize the promising potential of TENG technology for developing highly efficient and adaptable diagnostic systems for various respiratory conditions.

#### 3.1.2. TMSs for Hypoventilation Syndrome and Asthma Diagnosis

Peng et al. have made significant strides in this field by designing a modular electronic skin (e-skin) utilising TENG technology for real-time respiratory monitoring and obstructive sleep apnoea–hypopnoea syndrome (OSAHS) diagnosis (Figure 4c) [40]. Comprising 66 multilayer polyacrylonitrile and polyamide nanofibers as the contact pairs and deposited gold electrodes, the e-skin offers energy autonomy and accurate real-time respiratory monitoring. With its high-pressure sensitivity, good air permeability, and excellent working stability, the e-skin is a foundation for a self-powered diagnostic system that enables the real-time detection and severity evaluation of OSAHS to be achieved, ultimately improving sleep quality. Developed with a facile and low-cost electrospinning strategy, the e-skin possesses numerous micro-to-nano hierarchical porous structures that enhance contact electrification and facilitate thermal-moisture transfer. This allows the e-skin to achieve peak power density capable of powering hundreds of LEDs and charging various commercial capacitors under specific loading frequencies and applied force. With its immense potential for wearable medical electronics and personal healthcare monitoring, the e-skin emerges as a promising device for clinical applications.

Further advancements in TENG-based respiratory monitoring have been made by Yu et al., who developed a wearable point-of-care (POC) medical device specifically designed for respiratory diseases (Figure 4d) [41]. Comprising two modules, a self-switched triboelectric nanogenerator (SS-TENG), and a transdermal drug delivery system, the device offers a dynamic solution for continuous respiratory monitoring. The SS-TENG generates electrical energy from the wearer’s respiratory motions and operates in two modes: prevention and emergency. In prevention mode, the SS-TENG supplies continuous and stable power to the drug delivery system, which releases preventive asthma drugs. In emergency mode, which is activated by solid respiratory motion, the SS-TENG generates high-voltage pulses that trigger alarms and send emergency messages to monitoring terminals. The drug delivery system, which includes a micro-needle array, drug reservoir, and electrochemical actuator, releases drugs based on signals from the SS-TENG. The device’s modular architecture provides flexibility in switching between functions in response to respiratory motion and emits visual and wireless emergency alarms. This wearable SS-TENG device offers significant potential for the continuous monitoring and prevention of respiratory diseases, thereby reducing morbidity and mortality. The device’s feasibility and operability make it a valuable addition to POC prevention and the monitoring of respiratory diseases.

In conclusion, the TENG hypoventilation syndrome and asthma diagnostic system exemplifies modularity’s benefits in creating wearable devices for real-time respiratory monitoring and diagnosis. As demonstrated by Peng et al. and Yu et al., these modular devices show great potential for addressing respiratory health challenges and contributing to improved patient care and clinical outcomes with the integration of advanced materials and TENG technology.

#### 3.1.3. TMSs for Intelligent Mask Design

Lu et al.’s work exemplifies the value of modularity in the development of an intelligent facemask with a novel, structured respiratory sensing triboelectric nanogenerator (RSTENG) for respiratory monitoring (Figure 4e) [42]. The RSTENG comprises four distinct modules: a copper electrode layer, a polytetrafluoroethylene (PTFE) film layer, an aluminium foil layer, and a sponge layer. By integrating these modules, the facemask can monitor the breathing status and diagnose respiratory diseases, such as those caused by the COVID-19 pandemic. The modular design enhances the development of respiratory monitoring devices and their potential clinical applications. It achieves the integration of a breath-driven human–machine interface (HMI) system and an apnoea alarm system. These additional features empower users with disabilities to control small household appliances with breathing and provide timely alarms when breathing ceases, respectively.

In another example, He et al. designed a respiratory monitoring triboelectric nanogenerator (RM-TENG) using nanofibrous membranes for clinical use [43]. This innovative system also employs modularity, featuring three separate modules: the RM-TENG module, a data transmission module, and a data analysis module (Figure 4f). The RM-TENG module, comprising a polyacrylonitrile (PAN) and a PVDF electrospun nanofiber mat, generates electrical signals as the user breathes through the mask filter. These signals are wirelessly transmitted to a computer or smartphone by the data transmission module. The data analysis module processes and displays the information to obtain insights into various respiratory indices. The modular design of the RM-TENG system makes the accurate detection of the respiratory index over an extended period possible while maintaining excellent sensing stability and filtration efficiency. This makes it particularly suitable for providing valuable information about respiratory pathophysiology and analysing chronic obstructive pulmonary disease (COPD) severity.

Rajabi et al. further illustrated the benefits of modularity with the design of a respirator for respiratory monitoring [44]. This respirator consists of three modules: a TENG module, a power management module, and a wireless communication module (Figure 4g). The TENG module, made of a biocomposite film of diatom frustules and cellulose nanofibrils (CNFs), harvests energy from human breathing. Meanwhile, the power management module stores and regulates the harvested energy, and the wireless communication module transmits breathing signals to a smartphone app for monitoring. The modular design, combined with the use of biocompatible materials and high output performance, renders this system a promising candidate for clinical applications.

In conclusion, the modular face shield diagnostic system demonstrates the immense potential of modularity for respiratory monitoring. By incorporating advanced materials and customisable systems, it can provide accurate and reliable respiratory monitoring and diagnosis across various clinical applications. The examples provided by Lu et al., He et al., and Rajabi et al. showcase the effectiveness and versatility of modular systems in addressing respiratory health challenges, paving the way for the development of more advanced and user-friendly diagnostic tools in the future.

### 3.2. Cardiovascular System

#### 3.2.1. TMSs for Cardiac Real-Time Diagnosis

TENG cardiac real-time diagnosis represents a groundbreaking approach to heart activity monitoring, utilising TENGs for sensing and analysing various aspects of cardiovascular function [45,46,47,48]. This innovative method harnesses the power of modularity to develop compact, flexible, and biocompatible devices that boast enhanced functionality and seamless integration, revolutionising the field of cardiac monitoring and treatment (Table 3).

One remarkable example of modularity in TENG cardiac real-time diagnosis is the self-powered endocardial pressure sensor (SEPS) designed by Liu et al. Comprising four layers, this modular structure protects blood and moisture while enabling the real-time monitoring of endocardial pressure to be conducted (Figure 5a) [49]. The SEPS can be miniaturised, flexible, and integrated with a surgical catheter for minimally invasive implantation. Its excellent linearity and sensitivity allow cardiac arrhythmias such as ventricular fibrillation and ventricular premature contraction to be detected. By offering valuable information for heart failure patients, the SEPS holds significant clinical applications and potential for implantable healthcare monitoring, providing safe pressure sensing, diagnosis, and the monitoring of cardiovascular disease.

Further showcasing the importance of modularity, Zhao et al. designed a no-spacer triboelectric nanogenerator (NSTENG) for monitoring cardiovascular activity (Figure 5b) [50]. The system features three modules, including a copper electrode with a micro-/nanostructured surface to generate a gas gap by vaporising distilled water, an Ecoflex rubber layer for flexibility and biocompatibility, and a wireless mobile system for real-time signal transmission and display on a mobile phone. This eco-friendly in situ gap-generation method ensures biological safety and eliminates air contamination. The NSTENG can monitor normal and abnormal heart motion, providing additional information beyond traditional electrocardiograms. With its wearable and implantable nature, the NSTENG offers insights into developing bio-safe sensors and demonstrates its crucial role in cardiac monitoring by accurately detecting subtle cardiac motion that may go unnoticed in ECGs.

Hu et al. employed modular design principles to develop a superhydrophobic liquid–solid contact TENG (Figure 5c) [51]. This device features self-cleaning, self-adhesiveness, and high sensitivity, capturing and releasing triboelectrification energy upon droplet collisions or slipping on the superhydrophobic layer. As a sensor, the superhydrophobic TENG system comprises a signal acquisition circuit, a wireless transmission module, and a data processing module to collect, amplify, transmit, analyse, and display real-time signals. This technology has shown great potential for monitoring clinical drainage operations and intravenous injections in biomedical devices, as evidenced by prototypes such as a drainage bottle droplet sensor and an intelligent intravenous injection monitor. The superhydrophobic TENG also holds promise in diverse fields such as plasma separator devices, blood-repellent sensors, vascular stents, and heart valves.

In summary, incorporating modularity in TENG cardiac real-time diagnosis has facilitated the development of advanced, adaptable, and biocompatible devices capable of revolutionising the field of cardiovascular monitoring and treatment. By harnessing the advantages of modular design in devices such as the SEPS, the NSTENG, and the superhydrophobic TENG, researchers and medical professionals can access vital data and insights to enhance patient outcomes and drive innovation in cardiac care.

#### 3.2.2. TMSs for Pulse Real-Time Diagnosis

Pulse real-time diagnosis is an innovative approach in healthcare that utilises TENG technology to continuously monitor and evaluate cardiovascular health. TENGs can convert mechanical energy, such as the pulsatile motion of blood vessels, into electrical signals, which can then be analysed for various health indicators. By measuring these electrical signals in real time, healthcare professionals can obtain valuable insights into a patient’s heart rate, blood flow, and overall cardiovascular function, potentially allowing the early detection of anomalies and timely intervention for improved patient outcomes to be conducted [55,56,57,58,59,60]. When we use a modular design, we can separate the module that monitors pulse signals in real time and improve its comfort and durability with physical shape design. Moreover, we can place the energy collection module in a place that does not affect the everyday life of patients, improving the treatment quality of patients in all aspects.

Building on this concept, Wu et al. designed a multi-mode stretchable and wearable TENG device (MSW-TENG) using a modular design approach (Figure 5d) [52]. The device is composed of liquid metal and silicone materials and boasts stretchable and highly conductive properties. Its three adaptable working modes—contact separation, stretch, and press—make various applications possible. The MSW-TENG can monitor radial artery pulse signals, crucial for assessing cardiovascular health, by detecting changes in triboelectric charges induced by mechanical stimuli. With versatile applications in wearable electronics, this energy-harvesting, self-powered sensing device has tremendous potential for pulse monitoring and rehabilitation therapy.

Wang et al. designed a TENG system that utilises biocompatible polyvinyl alcohol (PVA) blends (Figure 5e) [53]. The system comprises three modules: a PVA blend film as the contact layer for the TENG, a copper film as the electrode layer for TENG, and a flexible substrate to support the PVA blend film and copper film. The PVA blend film can be tailored with various fillers, such as gelatine, HCl/NaOH, or KCl/NaCl, to adjust the dielectric constant and triboelectric performance. TENG technology efficiently transforms environmental and mechanical energy into electrical power, enabling self-powered health diagnostics and therapeutics to be conducted. The article examines how molecular and ionic fillers affect the PVA blends’ triboelectric performance by systematically engineering and characterising PVA blends at different structure levels. The optimized PVA-gelatine blended film demonstrates stable and robust electricity outputs and low detection limits of mechanical deformation, such as human pulse. PVA-blend-based TENG devices offer exciting potential for cost-effective human health monitoring and provide new strategies for designing future biocompatible TENGs.

Further exemplifying the importance of modular design, Wang et al. constructed a self-powered, real-time-sensing TENG vascular graft device (Figure 5f) [54]. This system features two main modules: an electrospun poly(3-hydroxybutyrate) (PHB) membrane as a positive tribo-material and an expanded polytetrafluoroethylene (ePTFE) membrane as a negative tribo-material. The PHB membrane serves as the substrate layer, while the ePTFE membrane functions as the packaging layer. Copper electrodes attached to these membranes collect and transfer induced charges. The device can detect hemodynamic conditions and support endothelial/vascular tissue regeneration. Its excellent pressure sensitivity during simulated pulsatile blood flow experiments suggests promise for the real-time monitoring and early detection of vascular disorders.

These interconnected examples demonstrate the significance of modularity in designing TENG devices for real-time pulse diagnosis, offering a foundation for future advancements in healthcare diagnostics and therapies.

## 4. Modular Design in Musculoskeletal Systems

### 4.1. Bone System

#### 4.1.1. TMSs for Bone Morphology Diagnosis

Real-time bone morphology diagnosis is a cutting-edge method involving the continuous observation and evaluation of bone structures and joint health [61]. This approach enables the prompt identification and diagnosis of issues to be achieved, including wear debris, joint and spinal motion, head movement during sleep, and bone healing processes. By embodying a modular design, real-time bone morphology diagnosis systems can be tailored to specific applications or user requirements (Table 4). This flexibility allows various components to be integrated, such as imaging devices, sensors, data processing units, and user interfaces, which can be easily combined or interchanged to create a customised diagnostic solution. Modularisation also facilitates seamless integration with other healthcare technologies and systems, enhancing the overall functionality and efficiency of the diagnostic process.

Li et al. designed a self-powered wireless sensor network based on TENGs for clinical applications, composed of four modules: energy harvesting, power management, sensing, and wireless communication (Figure 6a) [62]. The authors developed a high-sensitivity stretch sensing device that detects joint and spinal motions with excellent robustness and minimum resolution. This lightweight, precise, and durable stretch sensor is suitable for spinal monitoring and mitigating the risk of long-term posture-induced diseases.

Kou et al. designed a bright pillow utilising a flexible and breathable triboelectric nanogenerator (FB-TENG) pressure sensor array to monitor head movement during sleep (Figure 6b) [63]. Consisting of three modules, i.e., FB-TENG, data processing, and display, the pillow provides a non-invasive, comfortable solution for sleep monitoring with broad applications in personal healthcare, such as the real-time monitoring of brain diseases and cervical spondylosis. This modular design results in a self-powered, sensitive, low-cost, and comfortable device with practical clinical applications.

Liu et al. developed a self-powered, modular artificial joint wear debris sensor using a TENG [64]. This sensor converts mechanical energy from sliding movement into electrical signals, making wear debris detection possible. The innovative sensor is highly relevant for clinical applications, as it allows the timely and in situ detection of wear debris in artificial joint replacements to be achieved (Figure 6c). By preventing further joint deterioration and providing a basis for lifetime diagnosis, this TENG-based sensor represents a significant advancement in detecting wear debris in artificial joints, expanding the application of TENGs in biomedical sensors and intelligent healthcare.

Kaveh Barri and colleagues developed an innovative class of intelligent medical implants with advanced features, consisting of two primary modules: a TENG with an auxetic microstructure and a wireless communication module (Figure 6d) [65]. Without radiographic imaging, these self-aware metamaterial implants can diagnose bone healing processes and detect various spinal fusion levels using continuous stability and load-sharing measurements. This technology has the potential to revolutionise implantable devices and achieve superior surgical outcomes, further highlighting the significance of modularity in the real-time diagnosis and monitoring of bone morphology.

In conclusion, the modular approach in real-time bone morphology diagnosis has been demonstrated across various studies, highlighting its significance in the field. The research studies by Liu et al., Li et al., Kou et al., and Barri and colleagues all showcase the advantages of modularity in addressing different aspects of bone and joint health, sleep monitoring, and implantable devices, ultimately contributing to advancements in intelligent healthcare solutions.

#### 4.1.2. TENGs for Bone Repair

TENG bone repair, a groundbreaking method for promoting bone regeneration, employs TENGs to convert human body movement into electrical stimulation. This process activates cellular mechanisms crucial to bone healing. By harnessing TENG technology in bone repair, this non-invasive, self-powered, and energy-efficient approach presents a promising alternative to conventional bone healing techniques. It can potentially provide more effective and tailored treatment strategies in orthopaedics and regenerative medicine [68].

Wang et al. have pioneered a wearable pulsed triboelectric nanogenerator (WP-TENG) that leverages this technology to facilitate bone repair with the mechanosensation of Piezo1 (Figure 6e) [66]. Comprising multiple modules, the device generates stable pulsed electrical stimulation powered by human movement. At its peak value of 30 µA, the WP-TENG rejuvenates aged bone marrow mesenchymal stem cells (BMSCs), enhances their osteogenic differentiation, and promotes human umbilical vein endothelial cell tube formation. The study delves into the signal transduction mechanism for rejuvenating aged BMSCs and the theoretical basis for bone regeneration using triboelectric stimulation, offering valuable insights into the WP-TENG’s potential application in treating elderly patients.

In a complementary study, Yao et al. designed a device consisting of two modules: a TENG to generate electricity and a pair of dressing electrodes for direct electrostimulation at the fracture site (Figure 6f) [67]. This device provides closed-loop biofeedback therapy by generating electrostimulation signals exclusively from appropriate body motions, thus accelerating bone regeneration and maturation. The optimised electric field activates growth factors that regulate the bone microenvironment, fostering bone formation and remodelling. Remarkably, the device is battery-free, self-responsive, and requires no surgical removal after completing the biomedical intervention. By overcoming the complexities of equipment operation and stimulation implementation in clinical settings, this device makes biofeedback electrostimulation therapy readily accessible to patients. Clinical tests on rats demonstrated statistically significant improvements in mineral density and flexural strength, leading to effective bone fracture healing in as little as six weeks, compared with conventional therapy.

In both examples, the modularity of TENG bone repair devices enables enhanced performance and adaptability to be achieved, showcasing the potential of this innovative approach to bone regeneration.

### 4.2. Neuromuscular System

#### 4.2.1. TMSs for Motion System

The TENG-based sensing system for motion detection refers to an advanced technology that employs TENGs to monitor and analyse movement-related parameters [69]. TENGs can convert mechanical energy, such as vibrations or displacements, into electrical signals that can be analysed to provide valuable insights into motion patterns [70]. By embodying a modular design, we can produce systems with higher sensitivity and stability for motion diagnosis, such as combining the original TENG module with a wireless communication module filter module, which can create a customised motion detection solution. It enhances the overall functionality and efficiency of the motion detection process.

Yang et al. developed a self-powered medical nursing HMI system using modular design techniques (Figure 7a) [71]. Their system comprises three modules: a TENG device that transforms mechanical stimuli into electrical signals and can be attached to any joint of the body; a multichannel signal processing and encoding module for collecting and transmitting electrical signals from the TENG device; and a human–machine interface module that provides real-time feedback and assistance to patients and medical staff. The system integrates a TENG with a transparent, flexible–stretchable PVA/PA hydrogel as the electrode material and a single-electrode-mode sandwich structure using Ecoflex for sealing and triboelectric purposes. Moreover, a multichannel electrometer acquisition board is incorporated for processing and encoding electrical signals. This device offers real-time assistance by transmitting distress calls through gentle finger bending during diagnosis and can be integrated into any joint of the body as an active, self-powered flexible–stretchable triboelectric sensor, harvesting mechanical energy from ubiquitous motions. This study underscores the potential application of self-powered triboelectric sensors in the medical HMI.

Building on this, Li et al. designed a bioinspired sweat-resistant wearable triboelectric nanogenerator (BSRW-TENG) for movement monitoring during exercise (Figure 7b) [72]. The BSRW-TENG consists of two superhydrophobic and self-cleaning triboelectric layers (elastic resin and polydimethylsiloxane (PDMS)) with hierarchical micro-/nanostructures replicated from the lotus leaf. These micro-/nanostructures improve the electrical performance of the BSRW-TENG, resulting in a two-fold increase compared with the flat TENG. Furthermore, the BSRW-TENG boasts excellent contamination and humidity-resistant properties, contributing to sweat resistance. The device can withstand extreme conditions, including complete surface contamination and ultra-humid water spraying, and successfully monitors various exercise movements, such as dumbbell bicep curls, leg curls, and running, with stable performance before and after sweating. The BSRW-TENG demonstrates the importance of modular design in clinical applications, holding significant potential for low-cost personal exercise monitoring and athlete training analysis.

In a separate study, Yuan et al. introduced a multifaceted and integrated sliding system comprising two modules: a capacitive sensor and a TENG-based sensor (Figure 7c) [73]. The capacitive sensor is responsible for detecting compressive stress and deformation, while the TENG-based sensor ascertains the displacement and velocity of the contacting object. This modular design allows the device to register a broad spectrum of physical changes using electrical signals, guaranteeing accurate gripping control and averting unforeseen slipping and damage during the flexible gripping process. The TENG’s active attributes and self-powering capabilities render it an adaptable technology suitable for various applications, such as touch screens and electronic skins.

These advancements highlight the crucial role of modularity in fostering the development of inventive and efficacious diagnostic systems within the healthcare sector.

#### 4.2.2. TMSs for Parkinson’s Diagnosis

One such example is the modular system designed by Kim et al. (Figure 7d) [74]. It combines a highly stretchable and self-healable TENG for energy harvesting and tremor sensing with two distinct modules: a catechol–chitosan–diatom hydrogel (CCDHG) electrode module and an M-shaped Kapton film module. The biocompatible and eco-friendly CCDHG electrode module offers high stretchability, self-healing ability, and conductivity. In contrast, the M-shaped Kapton film module enhances the contact area and sensitivity of the TENG, acting as a tremor sensor to detect low-frequency vibrations from the human body. This self-powered tremor sensor can diagnose Parkinson’s disease by measuring the low-frequency vibrational motion of patients and has potential applications in biomedical health monitoring, intelligent e-skins, soft robotics, and wearable bioelectronics.

In another example, Wang et al. (Figure 7e) [75] developed a device consisting of two primary modules: a flexible strain sensor made from graphene oxide–polyacrylamide (GO-PAM) hydrogels and a data processing module. This system is designed for clinical use, particularly for identifying Parkinson’s disease and hemiplegia. The self-powered strain sensor, which relies on GO-PAM hydrogels, exhibits exceptional performance in detecting even the slightest human movements, including gait. The in-shoe wearable monitoring system leverages an artificial neural network algorithm to achieve impressive recognition accuracy for human daily-life speed and pathological rate, providing a convenient solution for various medical applications, such as early diagnosis, rehabilitation evaluation, and treatment of patients.

These modular approaches to real-time Parkinson’s disease diagnosis demonstrate the benefits of integrating multiple components to optimise system performance and make a wide range of applications in the medical field possible.

**Table 4 sensors-23-04194-t004:** Summary of TENGs in the musculoskeletal system.

Date	Position	Size (cm^2^)	Materials	Energy Source	Output	Application
2021 [62]	Wearable	3 × 3	Copper, Kapton	Vibration	80 V	Bone morphology diagnosis
2022 [63]	Wearable	2 × 2	FEP	Movement	2.5 V	Bone morphology diagnosis
2021 [64]	Wearable	None	PE/NI	Micro-vibration	98 V	Bone morphology diagnosis
2022 [65]	Wearable	None	PVA	Movement	9.2 V	Bone morphology diagnosis
2022 [66]	Bone	None	PTFE	Vibration	30 μA	Bone repair
2021 [67]	Bone	3.5 × 1.5	PLGA/Mg	Vibration	4.5 V	Bone repair
2022 [71]	Wearable	2 × 4.5	PVA/PA	Movement	1.33 W·m^2^	Motion sensing
2022 [72]	Wearable	2.8 × 2	PDMS	Movement	200 V	Motion sensing
2020 [73]	Wearable	None	FEP	Movement	150 nA	Motion sensing
2021 [74]	Wearable	3 × 3	CCDHG	Movement	110 V	Parkinson’s diagnosis
2022 [75]	Wearable	5 × 5	GO-PAM	Movement	26 mW	Parkinson’s diagnosis

## 5. Modular Design in Bacteria Diagnosis and Sterilization

### 5.1. TMSs for Gram-Positive Bacterial Diagnosis

In clinical application settings, the real-time diagnosis of Gram-positive bacteria is crucial to determining the appropriate treatment for patients suffering from bacterial infections (Table 5) [76]. The early and accurate identification of these pathogens helps medical professionals make informed decisions regarding antibiotic selection, reducing the risk of complications and improving patient outcomes [77]. Additionally, real-time diagnosis assists in preventing and controlling nosocomial infections, ensuring a safer healthcare environment [78].

One example of modularity in action is the self-powered biosensing system developed by Wang et al. for detecting Gram-positive bacteria in water samples (Figure 8a) [79]. This system consists of three main modules: a TENG module providing a stable voltage signal source, a sensor module with vancomycin-modified indium tin oxide glass and guanidine-functionalized multi-walled carbon nanotubes for capturing and amplifying bacterial signals, and a warning module that converts voltage signals into visual indicators using a Labview-based program. By leveraging the modular design, this system effectively detects Gram-positive bacteria in various applications, such as environmental pollution, iatrogenic diseases, and microbiological corrosion.

In another instance, Zhou et al. created a modular micro-biosensor system designed to quickly and sensitively detect Gram-negative bacteria, including sulphate-reducing bacteria (Figure 8b) [80]. This system comprises four distinct modules: ITO-ConA, for bacterial capture; CNT-ConA, for signal amplification and resistance reduction; a TENG module, for stable power supply; and an alarm module, for the direct display of detection results. The modular design of this micro-biosensor system ensures its portability, specificity, and stability, making it a valuable tool for the specific and sensitive detection of bacteria in clinical settings.

In both examples, the modular approach enables efficient, reliable, and user-friendly diagnostic systems that can be readily applied to various fields to be developed, highlighting the importance of modularity in the design of Gram-positive bacterial real-time diagnosis systems.

### 5.2. TENGs for Gram-Positive Bacterial Sterilization

The real-time sterilisation of Gram-positive bacteria exemplifies a pioneering approach for rapidly and proficiently eliminating detrimental microbes [83]. This avant-garde method plays a vital role in numerous industries and public health endeavours, as it aids in upholding hygiene, curbing the transmission of infections, and guaranteeing product safety. The foremost merit of real-time sterilisation technology lies in its modularity, which enables distinct yet interrelated components to be amalgamated to establish a cohesive, self-powered system. By incorporating specialised modules, these sterilisation systems can attain unparalleled levels of efficiency while mitigating environmental impact. For instance, one application might pair a piezoelectric energy-harvesting module with an advanced sterilisation module, working together to eliminate pathogenic bacteria. Zhang et al. have presented a modular device consisting of two key components: a TENG and a nanowire electrode array (NEA) (Figure 8c) [81]. The TENG module generates electricity to power the NEA module, sterilising urine by irreversibly electroporating pathogens using high-voltage pulsed electric fields. The resulting TENG-driven NEA (T-NEA) system demonstrates exceptional sterilisation efficiency, over 99.9999%, in synthetic urine contaminated with various bacterial strains. Moreover, the T-NEA system effectively degrades organic components in urine using radical oxygen species generated during its operation. This modular design creates a self-powered, eco-friendly system capable of eliminating harmful bacteria and pollutants from urine while preventing the production of toxic by-products.

Another example involves an energy-harvesting module that generates the requisite power for a sterilisation module employing a targeted mechanism, such as ultraviolet radiation. Chen et al. have developed an innovative self-powered antifouling UVC pipeline steriliser using a modular design approach to address microbial fouling and corrosion (Figure 8d) [82]. The steriliser comprises a modified soft-contact freestanding rotary triboelectric nanogenerator (MFR-TENG) and a UVC lamp module. The MFR-TENG module provides high-voltage output, harnessed to excite the UVC lamp module, generating ultraviolet radiation for sterilisation. This approach offers an efficient, cost-effective, and reliable solution to microbial fouling. The self-powered antifouling UVC pipeline steriliser demonstrates superior antimicrobial ability against typical corrosive bacteria, such as Halomonas titanicae and Pseudomonas aeruginosa. This modular design is efficient and safe as an eco-friendly option for water circulation systems.

Researchers and industry professionals can develop innovative, cost-effective, and environmentally responsible strategies to combat microbial contamination and protect public health by leveraging modularity in the real-time sterilisation of Gram-positive bacteria.

## 6. Conclusions and Prospect

### 6.1. TENGs in Energy Collection

#### 6.1.1. Power Output

To advance the power output and conversion efficiency of TENG devices, it is crucial to optimise material properties, device geometry, and electrode design, ensuring that the harvested energy is adequate for the target application. Material selection plays a vital role in enhancing the performance of TENG devices, with a focus on selecting materials that possess high triboelectric properties and mechanical solid characteristics. Additionally, researchers should investigate new materials or material combinations to further improve charge generation and transfer efficiency, potentially leading to groundbreaking innovations in the field. Regarding device structure and design, it is essential to optimise these aspects to maximise the contact area and relative motion between the triboelectric layers. This can be achieved by experimenting with various design configurations, such as vertical contact separation, lateral sliding, single-electrode, or freestanding mode, ultimately discovering the most effective setup for each application.

#### 6.1.2. Durability and Reliability

Selecting reliable raw materials is vital to improving stability and durability, ensuring devices can withstand various operating conditions while maintaining their performance. Therefore, researchers can focus on materials with high frictional electrical and solid mechanical properties to provide stronger elasticity and longevity to TENG equipment. Enhancing TENG stability in harsh environments is another essential consideration. We can choose protective coatings that can help safeguard devices from extreme temperatures, humidity, and wear, ensuring consistent performance even under challenging conditions.

#### 6.1.3. Multifunctionality

Researchers can choose to broaden the functionality of TENGs as a primary focus in the future. For example, integrating TENGs with other energy collection technologies, such as solar cells or thermoelectric generators, could give rise to hybrid systems capable of tapping into multiple energy sources, thereby enhancing overall energy output. This synergistic approach would maximize the benefits of each technology, potentially revolutionising the energy-harvesting landscape and contributing to a more sustainable future. Additionally, improving TENG stability using multi-module integration can further contribute to their durability and reliability. By utilizing interconnected TENG modules, devices can better handle the stresses and strains of different applications, distributing energy harvesting more evenly and improving overall system resilience.

### 6.2. TENGs in Sensing Systems

#### 6.2.1. Sensitivity and Signal Quality

In the pursuit of enhanced medical diagnostics, future research should concentrate on discovering innovative materials and designs to augment the sensitivity of TENG-based sensors. This entails directing research efforts toward synthesizing stable and sensitive new materials for creating sensors capable of detecting the most subtle changes in pressure, strain, or motion across various medical applications. Equally significant is the reduction in noise, which can be achieved by designing effective filter circuits to optimize signal quality.

#### 6.2.2. Biocompatibility and Comfort

As the demand for non-invasive and comfortable diagnostic tools increases, researchers must investigate novel non-toxic and safe materials for direct skin contact or implantation. This can be accomplished by selecting biocompatible raw materials for TENG fabrication. Concurrently, it is essential to design lightweight, flexible, comfortable, and unobtrusive sensors that seamlessly conform to the body’s contours, minimising the impact of the device on patients’ daily lives.

#### 6.2.3. Wireless Communication

Incorporating wireless communication capabilities into TENG devices is crucial to achieving real-time data transmission and remote monitoring. This advancement could expedite diagnostics and empower healthcare providers to make more informed decisions. Alongside wireless communication, developing advanced algorithms for processing and analysing collected data is critical. These algorithms could facilitate extracting meaningful information and identifying potential abnormalities or patterns associated with various disorders or diseases, ultimately enhancing the accuracy of medical diagnoses.

#### 6.2.4. Wearability and Durability

The effectiveness of TENG devices as continuous monitoring tools hinges on their wearability and durability. Researchers should focus on designing forms or novel materials that can withstand extended wear and tear while maintaining wearing comfort. Moreover, it is vital to investigate materials and fabrication technologies that can endure daily wear and environmental factors such as sweat, moisture, and temperature fluctuations. This could improve the stability of TENG devices in harsh environments. By addressing these factors, researchers can ensure the long-lasting performance and reliability of TENG devices in clinical diagnosis, positioning TENG-based equipment as a sustainable solution for the future of medical diagnostics.

### 6.3. TENGs in Bacterial Clinical Diagnosis and Sterilization

#### 6.3.1. Sensitivity and Selectivity

To improve the sensitivity and selectivity of TENG-based devices, researchers should focus on optimizing the surface functionalisation process. This can be achieved by utilising specific recognition elements, such as antibodies, aptamers, or molecularly imprinted polymers, which target particular bacterial strains or biomarkers. By doing so, TENG devices could detect the presence of specific substances or pathogens with greater accuracy and precision. This enhanced sensitivity and selectivity could ultimately lead to more reliable and accurate diagnostic tools that can aid healthcare professionals in making better-informed treatment decisions.

#### 6.3.2. Scalability and Integration

A crucial aspect of advancing TENG-based devices is the development of scalable fabrication processes. This would enable the large-scale production of TENG devices to be achieved, making them more accessible and cost-effective. Additionally, researchers should work towards ensuring that these devices can be easily integrated with existing diagnostic and sterilization systems, streamlining their implementation within healthcare facilities. TENG devices can become a mainstream solution for various diagnostic and sterilization applications by focusing on scalability and integration, contributing to a more efficient and effective healthcare system.

#### 6.3.3. Power Output and Efficiency

The power output and conversion efficiency of TENGs play a pivotal role in their overall performance. Researchers should concentrate on optimizing the material, structure, and design of TENG devices to enhance their power output and conversion efficiency. By achieving this, TENG devices can generate sufficient power to effectively operate diagnostic and sterilization equipment without compromising performance. Increased power output and efficiency could not only improve the functionality of TENG devices but also contribute to the development of more energy-efficient and sustainable healthcare solutions.

#### 6.3.4. Biocompatibility and Safety

In clinical settings, it is of utmost importance that TENG devices are made using biocompatible and non-toxic materials. This ensures the safety of patients and healthcare professionals, mainly when TENG devices come into direct contact with biological samples or are used on patients. Researchers should prioritize the selection of biocompatible materials and continuously assess the safety of these devices throughout their development process. By doing so, TENG devices can be confidently utilized in various healthcare applications, providing safe and effective solutions for diagnostics and sterilization.

## Figures and Tables

**Figure 1 sensors-23-04194-f001:**
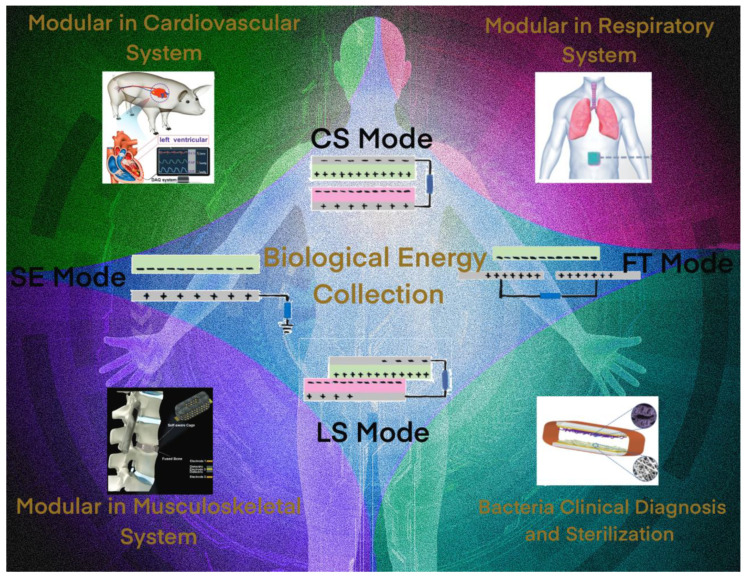
Modules in TENG sensors.

**Figure 2 sensors-23-04194-f002:**
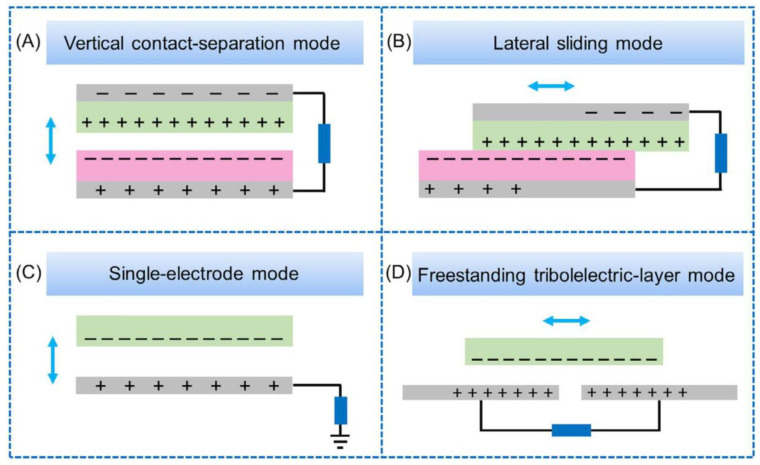
The following four modes are commonly used in TENGs: (**A**) Vertical contact–separation mode. (**B**) Lateral sliding mode. (**C**) Single-electrode mode. (**D**) Freestanding triboelectric layer mode. Reproduced with permission from [1] (MDPI, 2022).

**Figure 3 sensors-23-04194-f003:**
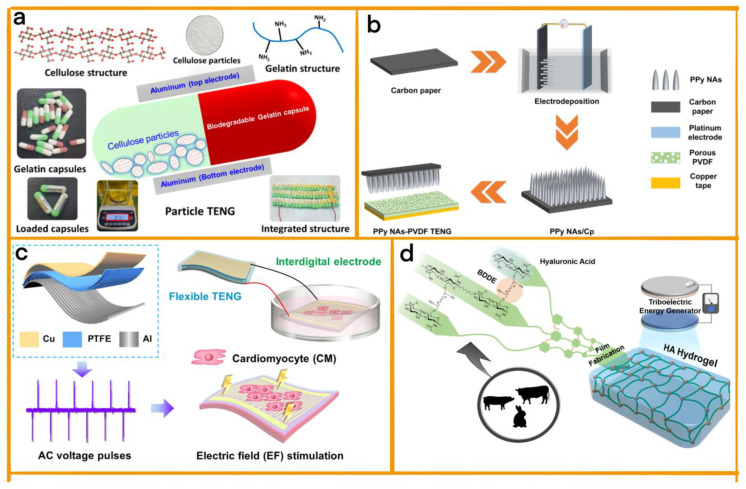
(**a**) A design scenario and schematic illustration of rapidly degradable P-, featuring tiny cellulose particles and a gelatine capsule. The FESEM image depicts the surface morphology [32] (Elsevier, 2022). (**b**) Schematic illustration of the fabrication process for PPy-PVDF TENG: Three-dimensional PPy NAs are deposited on carbon paper using electrochemical deposition and combined with a porous PVDF film to construct the PPy-PVDF TENG [33] (ACS, 2022). (**c**) A schematic diagram of the in vitro cardiomyocyte stimulation system demonstrates the electric field (EF) stimulation of cardiomyocytes generated by TENG on the interdigitated electrode. Scale bar: 300 μm [34] (Elsevier, 2022). (**d**) A schematic diagram presents a process chart for fabricating HA hydrogel film and its application in a TENG. The hydrogel films are made of hyaluronic acid extracted from mammals. An HA hydrogel film on ITO glass is integrated with a PTFE-coated Al film for TENG applications [35] (Elsevier, 2021).

**Figure 4 sensors-23-04194-f004:**
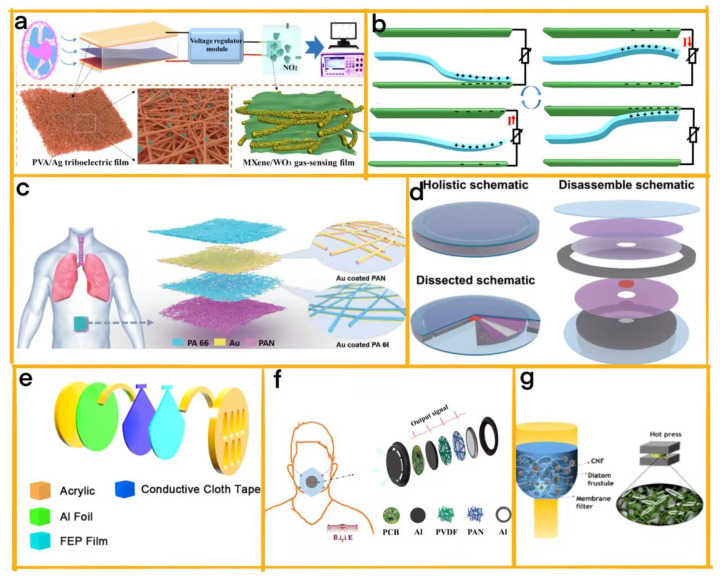
(**a**) A schematic illustration of a self-powered NO_2_ sensor driven by a TENG [38] (Elsevier, 2021). (**b**) The working mechanism of the prepared TENG, accompanied by SEM images [39] (Elsevier, 2022). (**c**) Structural design and working principle of the TENG-based SANES: (**a**) an application scenario showing the SANES attached to the abdomen for respiratory monitoring; (**b**) a schematic illustration of the SANES; (**c**) an enlarged view of the Au electrode layers coated on the surface of the PAN nanofiber film and PA 66 nanofiber film [40] (John Wiley and Sons, 2021). (**d**) A schematic of the SS-TENG, which is divided into three layers (fixed layer, connection layer, and motion layer) and includes four discal components and four annular components of varying sizes and materials [41] (Elsevier, 2023). (**e**) The detailed structure of the RS-TENG. (**b**) A photograph of a facemask assembled with an RS-TENG [42] (Elsevier, 2022). (**f**) A schematic illustration of the RM-TENG [43] (Elsevier, 2021). (**g**) A schematic illustration of the DF-CNF composite fabrication process [44] (ACS, 2021).

**Figure 5 sensors-23-04194-f005:**
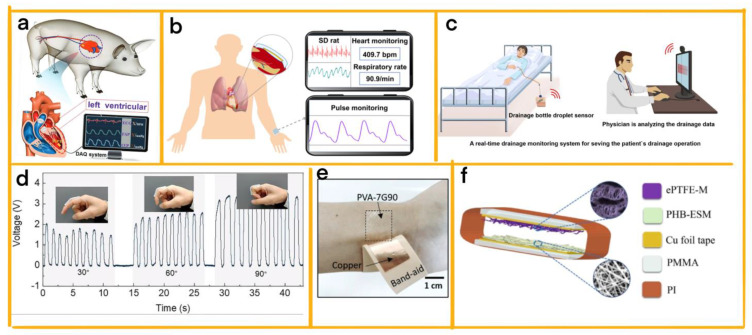
(**a**) A schematic diagram illustrating semaphore acquisition from the SEPS implanted into an adult Yorkshire swine’s heart [49] (John Wiley and Sons, 2019). (**b**) A schematic illustration depicting the NSTENG for monitoring the cardiovascular system [50] (Elsevier, 2021). (**c**) A schematic diagram of a drainage bottle droplet sensor designed for the real−time clinical monitoring of patient drainage operations [51] (ACS, 2020). (**d**) The output signal of the MSW−TENG under various finger bending angles during motion. The insets show the MSW−TENG attached to the finger [52] (Elsevier, 2022). (**e**) An optical image of the PVA−7G90 film attached to a human wrist [53] (John Wiley and Sons, 2020). (**f**) The structure of the ePTFE/PHB VC−TENG [54] (Elsevier, 2023).

**Figure 6 sensors-23-04194-f006:**
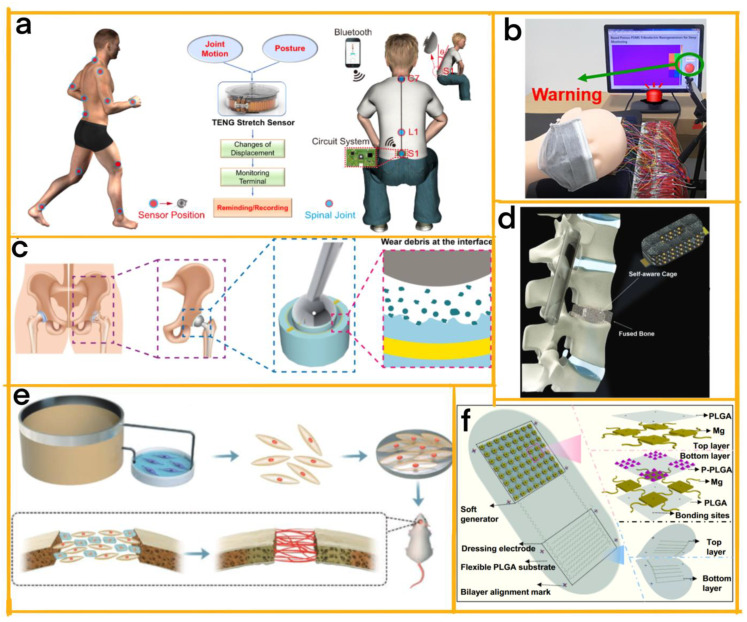
(**a**) An illustration of a slim, lightweight, wearable stretch sensor designed for full-body coverage to track joint and spinal movements, providing continuous wireless monitoring capabilities [62] (Nature, 2021). (**b**) A diagram demonstrating alarm system activation when the head reaches the edge column [63] (ACS, 2022). (**c**) A schematic representation of the wear debris sensor [64] (Elsevier, 2021). (**d**) A multifunctional nanogenerator interbody fusion cage featuring self-recovery, self-sensing, and energy-harvesting capabilities, designed for implantation during spinal fusion surgery [65] (John Wiley and Sons, 2022). (**e**) A schematic depiction of the bone defect repair and regeneration process in aged BMSCs activated by a WP-TENG in vivo [66] (John Wiley and Sons, 2022). (**f**) Schematic diagrams of the comprehensive FED structure (left) and a detailed illustration of the device components, essential materials, and multilayer structures (right) [67] (PNAS, 2021).

**Figure 7 sensors-23-04194-f007:**
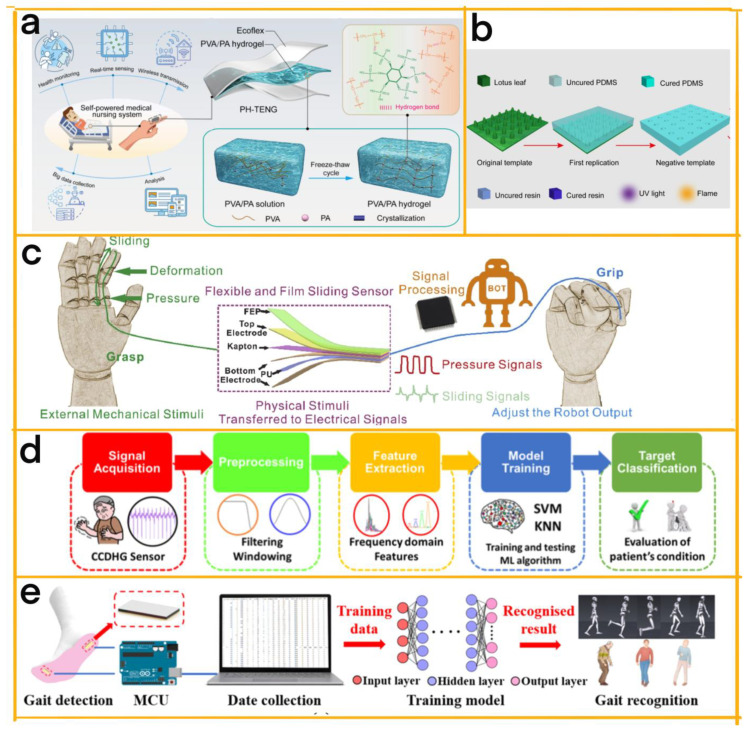
(**a**) An overview of the PH-TENG device assembled on the finger for health monitoring purposes [71] (Elsevier, 2022). (**b**) The replication process of the hierarchical micro-/nanostructure [72] (Elsevier, 2022). (**c**) A schematic representation of the flexible sliding sensor, designed for attachment to the finger surface to detect pressure and sliding signals [73] (Elsevier, 2020). (**d**) A flowchart illustrating the evaluation process for determining the condition of a Parkinson’s disease patient [74] (Elsevier, 2021). (**e**) A depiction of the human gait recognition process as conducted by the gait recognition model [75] (Elsevier, 2022).

**Figure 8 sensors-23-04194-f008:**
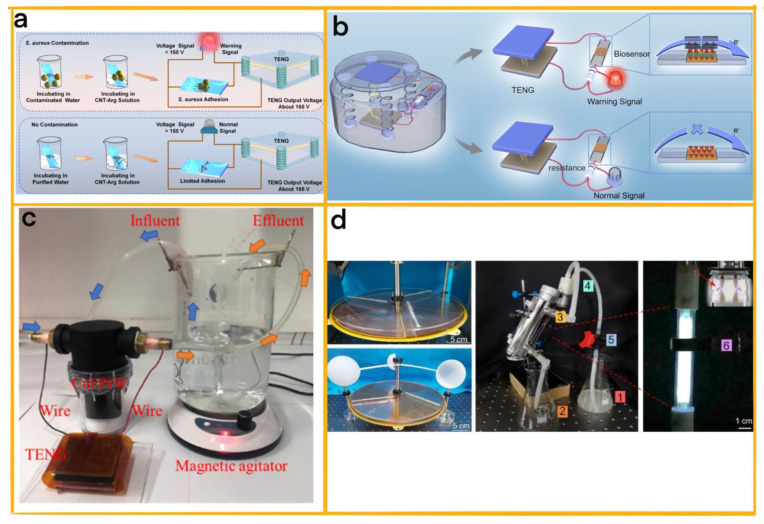
(**a**) A schematic illustration of the *S. aureus* detection process using a self−powered biosensing system, with the vertical contact–separation TENG as the voltage signal source in a liquid environment [79] (Elsevier, 2022). (**b**) The alarm is activated in the presence of bacteria, while it remains inactive when no or few bacteria are present [80] (Elsevier, 2022). (**c**) A photograph of the T−NEA urine sterilisation system [81] (Elsevier, 2022). (**d**) An image of the self−powered UVC pipeline steriliser [82] (Elsevier, 2022).

**Table 1 sensors-23-04194-t001:** Summary of TENG biological energy collection.

Date	Position	Size (cm^2^)	Materials	Energy Source	Output	Application
2022 [32]	Wearable	None	Cellulose particles	Particle vibration	70 μW	Electricity generation
2022 [33]	Wearable	2 × 2	3DPPyNAs	Movement	20.2 V	Electricity generation
2022 [34]	Heart	2 × 1.5	PTFE, AL	Heartbeat	20 V	Electricity generation
2021 [35]	Wearable	3 × 3	Hyaluronic acid	Vibration	20 V	Electricity generation

**Table 2 sensors-23-04194-t002:** Summary of TENGs in the respiratory system.

Date	Position	Size (cm^2^)	Key Materials	Energy Source	Output	Application
2021 [38]	None	4 × 8	PVA/Ag	Wind	530 V	NO_2_ sensor
2022 [39]	Nose	5 × 2	MXene	Breath	136 V	Formaldehyde sensor
2021 [40]	Nose	4 × 4	Polyamide 66	Breath	330 mW m^−2^	Apnoea–hypopnoea syndrome diagnosing
2023 [41]	Nose	None	PTFE	Breath	120 V	Respiratory diseases diagnosing
2022 [42]	Nose	None	FEP/AL	Breath	8 V	Respiratory sensing
2021 [43]	Nose	None	PVDF/PAN	Breath	110 mV	Respiratory sensing
2021 [44]	Nose	3.5 × 2.5	CNF	Breath	85.5 mW/m^2^	Respiratory sensing

**Table 3 sensors-23-04194-t003:** Summary of TENGs in the cardiovascular system.

Date	Position	Size (cm^2^)	Materials	Energy Source	Output	Application
2019 [49]	Heart	1 × 1.5	PTFE	Heartbeat	6.2 V	Heart disease sensing
2021 [50]	Pulse	2 × 2	Silicone rubber	Vibration	3.67 V	Cardiovascular sensing
2020 [51]	Pulse	None	PTFE	Vibration	90 nA	Cardiovascular sensing
2022 [52]	Wrist	5 × 5	Gallium/indium	Vibration	15 μW	Pulse real-time diagnosis
2020 [53]	Wrist	None	PVA	Vibration	29.2 nA	Pulse real-time diagnosis
2023 [54]	Wrist	5 × 5	PTFE	Vibration	280 μW	Pulse real-time diagnosis

**Table 5 sensors-23-04194-t005:** Summary of TENGs in bacterial diagnosis and sterilisation.

Date	Position	Size (cm^2^)	Materials	Energy Source	Output	Application
2022 [79]	Clinical	3.5 × 3.5	FEP/PVC	Vibration	165 V	Bacterial sensing
2022 [80]	Clinical	1 × 2	CNT-ConA	Vibration	160 V	Bacterial sensing
2022 [81]	Solution	None	PTFE	Vibration	230 V	Bacterial elimination
2022 [82]	Solution	None	PET/FEP	Vibration	26.5 V	Bacterial elimination

## Data Availability

Data are available upon request from the authors.

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
