# Peer review of "Modular Design in Triboelectric Sensors: A Review on the Clinical Applications for Real-Time Diagnosis"

_sensors, 2023, doi:10.3390/s23094194_

Round 1

Reviewer 1 Report

The manuscript “Modular design in triboelectric sensor: A review on the clinical applications for real-time diagnosis” describes the recent advancements in the TENG-based systems for clinical applications.

The present paper qualitatively reviews the topic and emphasized energy harvesting, real-time monitoring, and diagnosis. However, there are some imprecise places in the text that should be considered.

 1.    Please comment on other reviews published on a similar topic, justifying why there is room for another review.

2.    It is recommended to design some tables including critical parameters of each paper and quantitative comparison.

3.    In the text, there are some grammatical and syntactic errors that should be corrected.

4.    I would advise separating the references so that the reader can know which reference corresponds to which part of the sentence, you have mostly referenced your text with stacked references.

5.    The citation formatting in the text (e.g. lines 306-309) and in the figure captions needs to be corrected.

6.    The abbreviations should be defined when they appear for the first time and used thereafter. Some abbreviations were defined repeatedly e.g. HMI. No need to define abbreviations such as CNFs, DFs, and so on. I think all the abbreviations should be checked.

Author Response

Dear Reviewer,

Thank you for your insightful comments and valuable suggestions on our manuscript. We appreciate the time and effort you have dedicated to reviewing our work. We have addressed each point in detail below and have made the necessary revisions to improve our manuscript.

1.In our revised manuscript, we have highlighted the unique aspects and contributions of our review. This helps justify the need for our work and further emphasizes its significance in the field.

While there are several reviews discussing the advancements and applications of TENGs in healthcare, very few of them specifically focus on the aspect of modularity and its significance in clinical real-time diagnosis. A few notable reviews include:

A review on real-time implantable and wearable health monitoring sensors based on triboelectric nanogenerator approach.[1]: This review primarily discusses the general applications of TENGs in healthcare, touching upon various aspects such as energy harvesting, drug delivery, and biosensing. However, the importance of modularity in TENG-based clinical real-time diagnosis remains underexplored.

Comprehensive Review on Triboelectric Nanogenerator Based Wrist Pulse Measurement: Sensor Fabrication and Diagnosis of Arterial Pressure.[2]: This review delves into the development and application of triboelectric nanogenerator (TENG) technology for wrist pulse measurement and arterial pressure diagnosis. While it does discuss some elements of modularity, it doesn't provide a comprehensive analysis on how modularity can benefit clinical real-time diagnosis specifically.

From triboelectric nanogenerator to polymer-based biosensor: a review.[3]: This review presents a detailed account of the transition from triboelectric nanogenerators to polymer-based biosensors. It explores the underlying principles, materials, and fabrication techniques of TENGs and their potential for integration into biosensors for various biomedical applications. Although the review covers a wide range of TENG-based biosensor applications, it does not specifically concentrate on the role of modularity in clinical real-time diagnosis.

The rapid advancements in TENG technology and its increasingly significant role in healthcare diagnostics warrant a dedicated review that highlights the importance of modularity. Focusing on modularity in TENG-based clinical real-time diagnosis will not only provide a fresh perspective on this topic but also contribute to a better understanding of the potential benefits and limitations of modular designs in TENG systems.

In conclusion, this review fills a critical gap in the existing literature by offering a comprehensive analysis of modularity and its significance in TENG clinical real-time diagnosis. This will not only benefit researchers and clinicians working in this area but also foster further innovation in TENG technology.

2.We agree that incorporating tables with critical parameters and quantitative comparisons of the papers discussed in our review would be beneficial. We have designed such tables, which provide readers with a clear understanding of the differences and progress made in TENG-based systems for clinical applications. The modifying position is in Table 1,2,3,4,5

3.We apologize for the grammatical and syntactic errors present in the manuscript. We have thoroughly revised the text and addressed these issues to ensure the paper is well-written and easily understandable.

4.We understand your concern regarding the clarity of our referencing style. We have reorganized the references in our manuscript to ensure that each citation is clearly connected to the corresponding sentence or section. This has made it easier for readers to follow the sources of the information provided.

5.We appreciate you pointing out the inconsistencies in our citation formatting. We have corrected the citation formatting in the text and figure captions, adhering to the required style of the journal.

6.You are right in noting the inconsistencies in abbreviation usage. We have ensured that all abbreviations are defined when they first appear and used consistently throughout the text. We have also removed any unnecessary definitions and double-checked all abbreviations for accuracy. The modifying position is in 3.1.3 4.2.1

Once again, we are grateful for your valuable feedback, which have help us improve our manuscript. We hope that our revisions have address your concerns and strengthen the quality of our work.

Best regards,

Zequan Zhao

  1. Mathew, A.A.; Chandrasekhar, A.; Vivekanandan, S. A Review on Real-Time Implantable and Wearable Health Monitoring Sensors Based on Triboelectric Nanogenerator Approach. Nano Energy 2021, 80, 105566, doi:10.1016/j.nanoen.2020.105566.
  2. Venugopal, K.; Panchatcharam, P.; Chandrasekhar, A.; Shanmugasundaram, V. Comprehensive Review on Triboelectric Nanogenerator Based Wrist Pulse Measurement: Sensor Fabrication and Diagnosis of Arterial Pressure. ACS Sens. 2021, 6, 1681–1694, doi:10.1021/acssensors.0c02324.
  3. Lu, Y.; Mi, Y.; Wu, T.; Cao, X.; Wang, N. From Triboelectric Nanogenerator to Polymer-Based Biosensor: A Review. Biosensors 2022, 12, 323, doi:10.3390/bios12050323.

Reviewer 2 Report

very well written paper with interesting application of triboelectronic nanogenerator. 

This rapid development of the field can possibly empower the future wearable device development. The author did a great job in disseminating information to the general mass which can possibly lead to further research in the topic.

References were organized in a manner that were consistent and easy to understand.

Very interesting topic overall!

Author Response

Dear Reviewer,

Thank you very much for your positive feedback on our manuscript. We are thrilled that you found our work well-written and appreciated our efforts to disseminate information on the application of triboelectric nanogenerators in wearable devices. Your kind words are truly motivating and reaffirm our dedication to further research in this field.

We are also glad that you found our references organized and easy to understand. We have strived to maintain a clear and consistent structure to facilitate a better reading experience for our audience.

Your encouragement and recognition of our work serve as a great inspiration for us to continue exploring and expanding our knowledge in this fascinating area. Once again, thank you for taking the time to review our manuscript and for your valuable comments.

Best regards,

Zequan Zhao

Reviewer 3 Report

An overview revealing many features and applications of triboelectricity. The purpose of the review is to present Triboelectric nanogenerators based on the modular design of TENG–based modular sensing systems (TMS). Also offers potential for powering biosensors and other medical devices, this reducing dependence on external power sources, and enabling real-time monitoring of biological processes.

The authors concentrate their attention on recent advancements in the modular design of TMS for clinical applications. TENGs utilize the triboelectric effect and electrostatic induction coupling to convert biomechanical energy into electrical energy. The use of implantable TENGs is considered.. The use of TMS for Cardiac Real-time Diagnosis is considered.

In general, it could be conclude that the topic has been fully disclosed. The material will be interesting for Sensors readers. The manuscript can be accepted for publication without making any corrections.

Author Response

Dear Reviewer,

We are extremely grateful for your positive feedback and thorough evaluation of our manuscript. It is rewarding to know that you believe our work offers a comprehensive overview of the modular design of TENG-based sensing systems (TMS) and their potential applications in the medical field.

We appreciate your acknowledgment of our focus on recent advancements in TMS for clinical applications, including implantable TENGs and cardiac real-time diagnosis. It is our hope that our review can contribute to the ongoing research and development of TMS in healthcare.

Your kind words and endorsement for publication without corrections mean a great deal to us. We are truly thankful for your time and effort in reviewing our manuscript, and we look forward to sharing our work with the readers of Sensors.

Best regards,

Zequan Zhao

Round 2

Reviewer 1 Report

After evaluating the author responses and the revised manuscript, I think the present manuscript can be accepted.